# Establishment of the body condition score for adult female *Xenopus laevis*

**Leonie Tix** [1], **Lisa Ernst** [1], **Britta Bungardt** [1], **Steven R. Talbot** [2], **Gero Hilken** [3], **René H. Tolba** [1] *

1 Institute for Laboratory Animal Science & Experimental Surgery, Faculty of Medicine, RWTH Aachen University, Aachen, Germany, 2 Institute for Laboratory Animal Science, Hannover Medical School, Hannover, Germany, 3 Central Animal Laboratory, University Hospital Essen, University Duisburg-Essen, Essen, Germany

* rtolba@ukaachen.de

**Data Availability Statement:** All relevant data are within the manuscript, its Supporting Information files and under the link: https://github.com/mytalbot/frog_BCS_data.

## Abstract

The assessment of animals' health and nutritional status using a Body Condition Score (BCS) has become a common and reliable tool in lab-animal science. It enables a simple, semi-objective, and non-invasive assessment (palpation of osteal prominences and subcutaneous fat tissue) in routine examination of an animal. In mammals, the BCS classification contains 5 levels: A low score describes a poor nutritional condition (BCS 1–2). A BCS of 3 to 4 is considered optimum, whereas a high score (BCS = 5) is associated with obesity. While BCS are published for most common laboratory mammals, these assessment criteria are not directly applicable to clawed frogs (*Xenopus laevis*) due to their intracoelomic fat body instead of subcutaneous fat tissue. Therefore, this assessment tool is still missing for *Xenopus laevis*. The present study aimed to establish a species-specific BCS for clawed frogs in terms of housing refinement in lab-animal facilities. Accordingly, 62 adult female *Xenopus laevis* were weighed and sized. Further, the body contour was defined, classified, and assigned to BCS groups. A BCS 5 was associated with a mean body weight of 193.3 g (± 27.6 g), whereas a BCS 4 ranged at 163.1 g (±16.0 g). Animals with a BCS = 3 had an average body weight of 114.7 g (±16.7 g). A BCS = 2 was determined in 3 animals (103 g, 110 g, and 111 g). One animal had a BCS = 1 (83 g), equivalent to a humane endpoint. In conclusion, individual examination using the presented visual BCS provides a quick and easy assessment of the nutritional status and overall health of adult female *Xenopus laevis*. Due to their ectothermic nature and the associated special metabolic situation, it can be assumed that a BCS $\geq$ 3 is to be preferred for female *Xenopus laevis*. In addition, BCS assessment may indicate underlying subclinical health problems that require further diagnostic investigation.

## Introduction

*Xenopus laevis*, the Smooth or South African clawed frog, belongs to the genus *Xenopus* and the family Pipidae and is classified in the order Anura. There are numerous *Xenopus* species

**Funding:** The authors declare funding in part from the German Research Foundation (Deutsche Forschungsgemeinschaft—DFG; FOR-2591, TO 542/5-2, TO 542/6-2 to R.T.; BL 953/10-2, BL953/11-2 to A.B.) without the involvement of the funders in study design, data collection, data analysis, manuscript preparation or decision to publish.

**Competing interests:** The authors have declared that no competing interests exist.

and subspecies described. However, only the larger *Xenopus laevis* and the much smaller Western clawed frog (*Xenopus tropicalis*) are relevant for animal research [1].

The first described use of South African clawed frogs for experimental and testing purposes dates back many decades. It was quickly recognized that they were suitable experimental animals, and scientists started to enable and improve their husbandry and breeding. Since adult clawed frogs, in particular, are undemanding in terms of husbandry, they can be kept and bred for long periods with little effort [2–8]. Their use and application as a testing method for pregnancy until the 1960s earned the frogs the nickname "Apothekerfrosch" (pharmacy frog) in Germany. Since then, the unpretentious, purely aquatic amphibians have become integral to laboratory animal science [9]. For example, according to the number of animals reported in the annual German laboratory animal reports (Ordinance on the Reporting of Vertebrates or Cephalopods Used for Experimental Purposes or Vertebrates Used for Certain Other Purposes, VersTierMeldV 2013), the number of clawed frogs used in German laboratory animal facilities in 2020 was 10,486 [10].

The ability of these frogs to live purely aquatically, to lay hormone-induced repeatedly numerous eggs in the water (>10,000 eggs/female/year), the resistance and the size of the eggs made *Xenopus laevis* a preferred model organism in developmental physiology [11]. Spawning of the animals can be artificially induced by hormone stimulation (injections of human chorionic gonadotropin (hCG) into the dorsal lymph sac [12] or via tank water [13]). In addition, surgical removal of ovarian tissue by laparotomy or spreading are possible ways to obtain unfertilized eggs [1].

This causes the preferential interest in female *Xenopus laevis* within animal-based research. Furthermore, due to their high life expectancy (≥20 years) [14], the animals can be used for several years until the egg quality or egg quantity decreases [15] to the extent that the animals have to be excluded from experiments.

The usage of *Xenopus laevis* in research focuses on genetic material manipulation and the developmental processes in the eggs and the tadpoles. In addition, the study of physiological processes in the cell (e.g., nuclear pore proteins) is of interest [2, 16–18].

Despite their widespread use, determining these animals' health affection or suffering through chronic experiments or housing conditions is quite challenging. Unlike other laboratory animals, these animals do not have comparable facial expressions (see available grimace scales for mice, rats, pigs, sheep, monkeys, horses, etc. (15)). Their facial expressions cannot address the experimentally-related severity. Their physiology, strongly influenced by the housing conditions, is not comparable to that of mammals [19]. Due to their ectothermy, standard assessment parameters can be transferred to amphibians only with difficulties. Parameters such as body temperature, skin turgor, telemetry, behavioral testing, and recurrent hematological or serological examinations are inappropriate measures or need to be adjusted. Some of these parameters are potentially suitable for application to the clawed frog (blood tests, behavioral tests [20–22]) but are complex or still immature in their routine applicability.

In addition to experiments or housing conditions, infections and parasite infestations can weaken the animals. There are few reports on disease in clawed frogs [1, 6, 23]. International trade of the invasive South African clawed frog (*Xenopus laevis*), a subclinical carrier of the fungal pathogen *Batrachochytrium dendrobatis* (Bd), has been proposed as a primary means of the introduction of Bd into native, susceptible amphibian populations. The historical presence of *B. dendrobatides* in the indigenous African population of *Xenopus* is well documented [24, 25].

Moreover, the assessment of individual *Xenopus laevis* is complicated and highly variable by the following factors: often housed in large groups of animals per tank, close contact between animals (heap lying), the influence of water quality, temperature, and feeding factors on weight development and growth.

In general hygienic and species-appropriate husbandry conditions, adequate feeding, qualified personnel, and correct handling of the animals are assumed to be the essential prerequisites for a stable state of health and body condition of the animals. However, for this species, species-specific criteria are still needed to determine, assess, and evaluate possible severity in animal experiments. It must be possible to determine the severity of an individual specimen at the actual time and during the experiment, considering its species-specific anatomy, physiology, and behavior. In principle, the preparation of this severity assessment is required by law as a prospective assessment already in the experimental design (e.g., Annex VIII EU Directive 2010/63 or § 7 para. 1, no. 2, § 8 section. 1, no. 7 German Animal Welfare Act, § 17) [26, 27]. Here, animal experiments commonly use species and protocol-specific score sheets as assessment tools. In Germany, recommendations for severity assessment in animal experiments and explicitly for clawed frogs are made by the Committee for Animal Welfare Officers of the GV-SOLAS [28].

In general, different criteria are required for the assessment of an individual [29], e.g.:

- assess the animal with as little stress as possible

- identify and rank the expected signs of severity using predefined criteria

- can be applied by various personnel in the same way

- be able to record and depict the chronological course of severity that occur

- enable predictions of potential, expected, and subsequent severity

- enable documentation of progress controls in the case of treatment

- Definition of termination criteria or humane endpoints.

At the same time, the quantitative, qualitative, and temporal criteria to be investigated must be:

- adapted to the experimental design (here, e.g., surgical intervention with wound suture or injection for egg laying)

- chosen to recognize severity caused by changes in individual behavior or organ systems (e.g., skin changes, lethargy, emaciation).

For a proper severity assessment of *Xenopus laevis*, compliance with husbandry and hygiene regulations is indispensable. Likewise, a basic knowledge of physiology, anatomy and the biological activity pattern of the test animal needs to be assessed [30]. Here, the body condition score is another standard tool for severity or health assessment. This score determines the nutritional status based on palpable bone landmarks and subcutaneous fat depots and classifies animals into 5 nutritional levels, ranging from emaciated up to obese.

In other laboratory animal species, the use of those species-specific body condition scores (BCS) is already established (zebrafish, mouse, rat, rabbit, cats, dog, sheep, pig, NHP) [31–40], but there is still no BCS for clawed frogs. It should be noted that *Xenopus* do not have palpable subcutaneous adipose tissue, as their only fat body is intracoelomic [41]. Thus, standard examination criteria are difficult to transfer to the frog due to its particular anatomy. The score described here represents a simple approach for the experimental refinement of these animals within procedures.

The study aimed to establish a scoring system to assess frog condition using non-invasive and objective assessment criteria. This score can be used as an additional tool in the scoring process and routine animal check-ups, e.g., as an individual health status indicator. Further,

we investigated whether the classification to a BCS group depends on the body weight or the length of a *Xenopus laevis*.

Due to the experimental approaches of egg and oocyte harvesting by the research groups working with the animals assessed here, the present study used female adult *Xenopus laevis* animals to establish the body condition score.

## Materials and methods

### Animals and husbandry

The animals were bred by the commercial breeder Xenopus 1 (Xenopus-I Inc, Michigan, USA) as lab-breed animals for experimental purposes. They were ordered with a body size of at least 12 cm in length (nose to cloaca). This assumed that all animals were sexually mature and of adult age and could be used for egg-laying protocols after acclimation at the experimental site. However, no statement can be made about the age or the degree of relationship of the animals. The Governmental Animal Care and Use Committee approved the housing and use of all animals for scientific purposes (Reference No.: 81–02.04.2020.A107; 81–02.04.2019. A355; 81–02.04.2019.A356; Landesamt für Natur, Umwelt und Verbraucherschutz Recklinghausen, Nordrhein-Westfalen, Germany).

The animals used were kept in three different husbandry systems:

1. individual water tanks (length x width x height (LxWxH): 115x55x36 cm), no water conditioning; average water temperature 12˚C ±3˚C, n = 20 animals.

2. circulating, a semi-closed system with 4 tanks arranged in parallel (Aqua Schwarz GmbH, Göttingen, Germany; LxWxH: 99x50x21 cm); conditioning of water by mechanical filtration systems and UV radiation; average water temperature 20˚C ± 1˚C, n = 20 animals

3. circulating, a semi-closed system with 17 tanks arranged in parallel (Aqua Schwarz, Göttingen, Germany; LxWxH: 100x50x25 cm); conditioning of water by mechanical filtration systems, UV radiation, demineralization, salting, and activated carbon filtration; average water temperature 20˚C ± 1˚C, n = 22 animals.

All animals were marked for individual identification with numbered metal tags (mouse ear tag, ITEM: 56779, Stoelting Co., Ireland) inserted into the webbing of the lateral outer toes of the right hind leg.

A total of 62 female adult *Xenopus laevis* were assessed within this study. The examined animals were randomly selected, whereas at least 20 of each husbandry system were used. All animals were fed every working day with a formulated dry pellet diet (Ssniff Spezialdiäten GmbH, Soest, Germany, Alleinfuttermittel für Krallenfrösche, V7106-030, 5 mm extrudate, 3 pellets per animal). Feed that was not immediately ingested was removed after 30 minutes. Within one day (24h), 3–5% of the total water volume was replaced by fresh water by continuous rinsing. The day-night cycle was 12 hours each. The monitoring of the tank system and the animal population, as well as the corresponding documentation of the water quality, took place daily.

### Body Condition Scoring (BCS)

The presented BCS is based on established concepts of body condition scores of other species [31–34, 38, 39, 42], including *Xenopus laevis*- specific physical characteristics of its body. The classification is made in 5 ascending score levels (Figs 1 and 2), whereby BCS 1 is associated with a humane endpoint due to emaciation (Fig 3), and BCS 5 is regarded as a very well-conditioned physical state. The shape of the upper, lower, and lateral body silhouette, muscling and fleshing of the hind limbs, and visibility/protrusion of the thoracic and pelvic bone points are

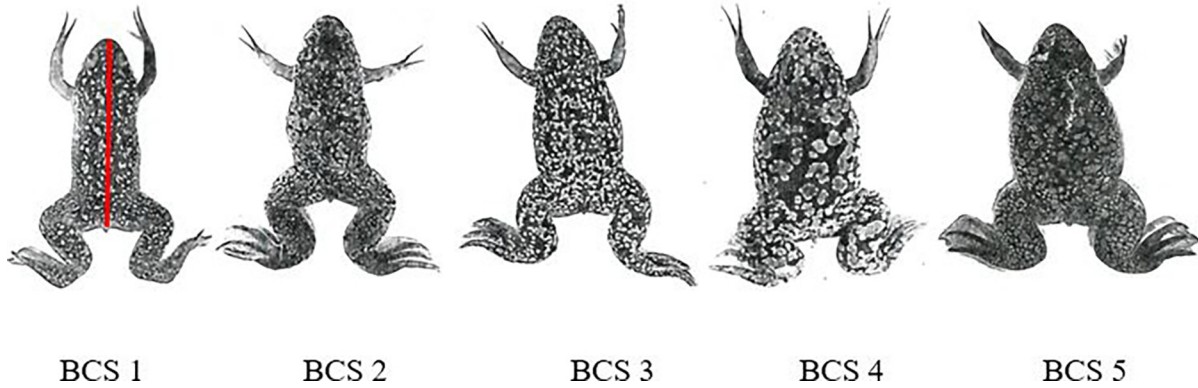

**Fig 1. Dorsal section of *Xenopus laevis* and assignment to BCS 1–5, red line in BCS1 indicates the nose-cloaca distance.**

considered in the evaluation. The following assessment criteria are distinguished and defined for the respective BCS levels (S1 Fig).

**BCS 1.** Frog emaciated (—): rectangular body shape; raised abdominal contour and flat dorsal contour; bony structures of thorax and pelvis are prominent in ventral view; skinny limbs firmly set off from trunk; humane endpoint reached.

**BCS 2.** Frog underconditioned (-): almost rectangular body shape with only slight curves left above the insertion of the posterior extremities; flat abdomen and dorsal contour; bone structures of the thorax can be adumbrated in ventral view; lean limbs set off from the trunk.

**BCS 3.** Frog moderately well-conditioned (+): slightly convex silhouette (also at the ventral and dorsal contour); limbs set off from trunk with early fleshing and pronounced musculature.

**BCS 4.** Frog well-conditioned (++): slightly pear-shaped body; abdominal and dorsal contour with unmistakable convex silhouette; muscular thighs.

**BCS 5.** Frog very well-conditioned (+++): pear-shaped body; upper and lower body contour with even, distinct convex curvature; fleshy, very muscular limbs.

## Body weight and metrics

To assess the animals, they were individually removed from their housing tank with a fishing net, placed in an empty bowl, and weighed. Then the body dimensions were determined with a flexible ruler (nose-cloaca length, NCL). Hereafter, the bowl was filled with water and scored and photographed from above and from the side. Here, the water was taken from the animals' housing tank to avoid distress for the animal.

## Statistics

Statistical analyses were performed in R (v4.0.3) [43]. Data were tested against the hypothesis of normal distribution using the Shapiro-Wilk test and were visually inspected with a QQplot. Normally distributed body weight data were analyzed in linear regression with treatment contrasts (BCS = 5 as the intercept level). BCS-class differences were represented as relative coefficient differences (planned contrasts) and were reported as estimates ($\beta$) with the corresponding standard errors (SE) as well as the p-values. Between BCS-class differences were analyzed with a one-way analysis of variance, followed by Tukey-Kramer post hoc tests to adjust for multiple comparisons. Post hoc test results were expressed as marginal means. In the nose-cloaca length (NCL) analysis, the body weight was analyzed as a fixed effect and for its interaction with BCS

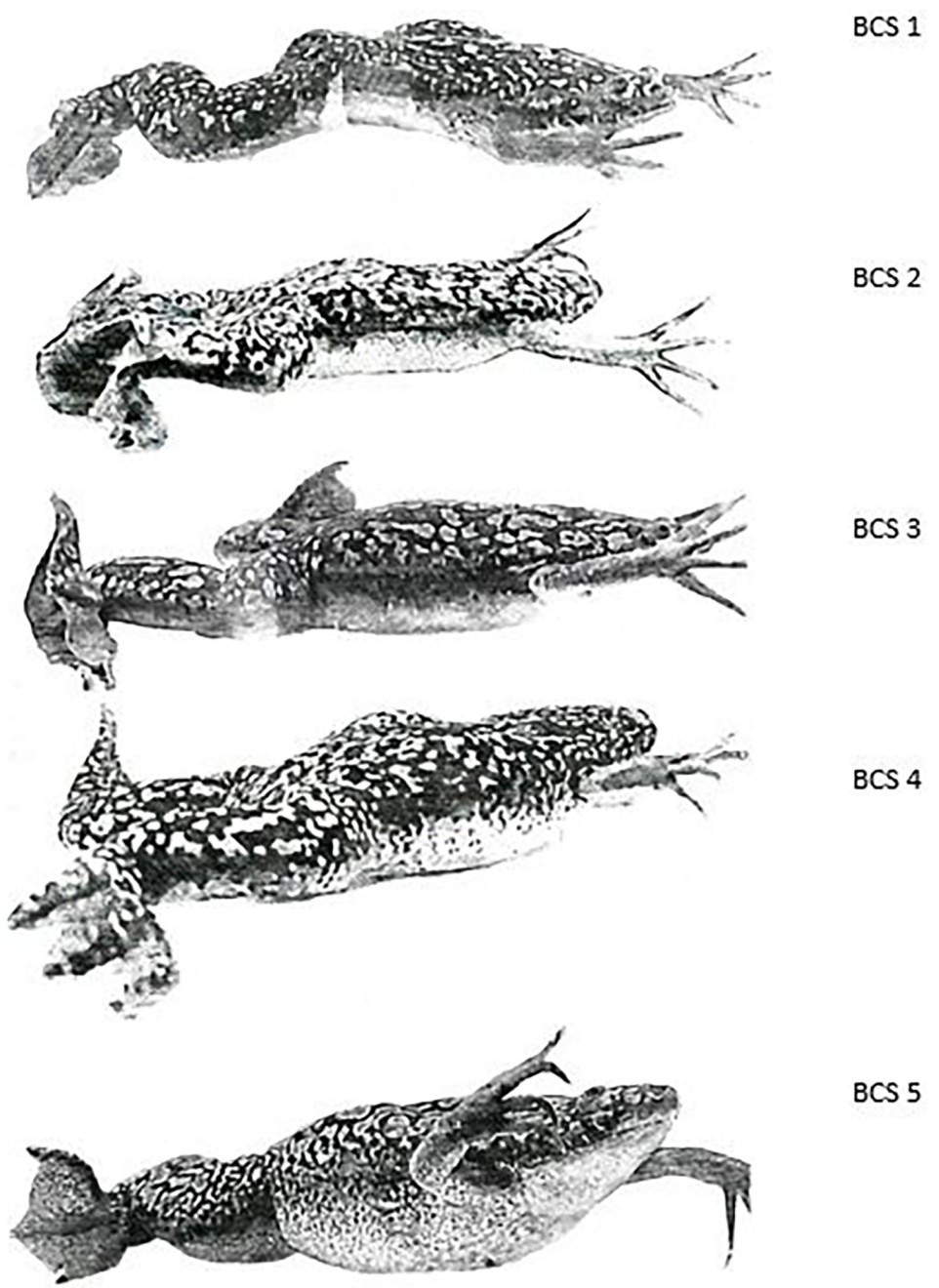

**Fig 2. Lateral section of *Xenopus laevis* and assignment to BCS 1–5.**

(bw:BCS). Both models were compared in a likelihood-ratio test using the lmtest package [44]. The following p-values were considered significant at the levels: $p<0.05$ (*), $p<0.01$ (**), and $p<0.001$ (***). Data were visualized with the ggplot2 package [45].

## Results

A total of 62 adult female *Xenopus laevis* were assessed within this study. While BCS 1 is accompanied by a humane endpoint and should not occur in an ideal healthy animal

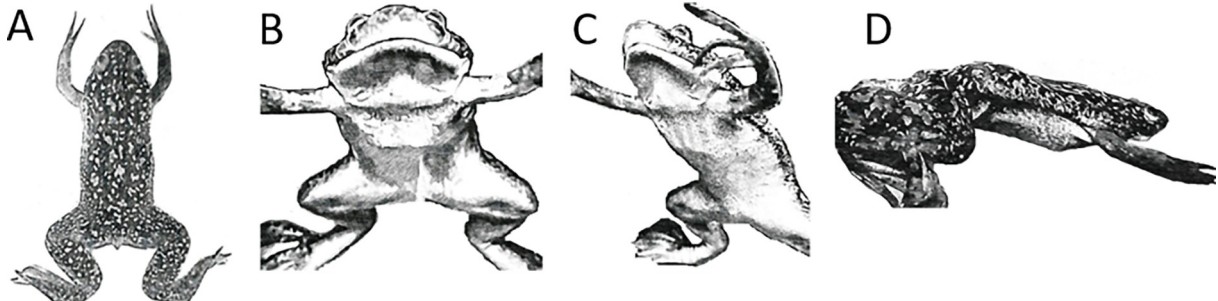

**Fig 3.** Close-ups pictures of BCS 1: A: dorsal view, B and C: ventral and latero-ventral view, D: lateral view. Humane endpoint: Frog emaciated (—): rectangular body shape; raised abdominal contour and flat dorsal contour; bony structures of the thorax as well as pelvis prominent in ventral view; lower side of the mouth with concave curvature (sunken); skinny limbs firmly set off from trunk; humane endpoint reached.

population, this score was observed once. The BCS 2 group comprised 3 frogs, and the BCS 3 group included 16 animals. In comparison, BCS groups associated with good up to optimal nutritional status were found in 18 (BCS4) and 24 (BCS5) animals. The absolute body weights of the animals ranged from 83 g to 235 g. The allocation to the respective BCS groups is shown in Table 1. The distribution of assessed BCS within the different housing conditions and their effects on BCS are provided in the supplemental materials (S1 File).

Normally distributed frog body weight data (W = 0.980, p-value = 0.42) were fit with a linear regression using the BCS as the independent variable. The BCS consisted of five-factor

**Table 1. Five classes of body condition score assigned by body weight (g).**

| BCS 1 | BCS 2 | BCS 3 | BCS 4 | BCS 5 |
|---|---|---|---|---|
| 83 | 111 | 112 | 161 | 173 |
|  | 100 | 146 | 158 | 172 |
|  | 103 | 138 | 119 | 178 |
|  |  | 151 | 147 | 213 |
|  |  | 156 | 167 | 140 |
|  |  | 134 | 172 | 211 |
|  |  | 134 | 146 | 226 |
|  |  | 172 | 190 | 217 |
|  |  | 153 | 150 | 190 |
|  |  | 150 | 174 | 223 |
|  |  | 118 | 157 | 189 |
|  |  | 167 | 171 | 197 |
|  |  | 137 | 181 | 181 |
|  |  | 144 | 166 | 163 |
|  |  | 131 | 172 | 235 |
|  |  | 171 | 173 | 229 |
|  |  |  | 179 | 150 |
|  |  |  | 153 | 227 |
|  |  |  |  | 234 |
|  |  |  |  | 156 |
|  |  |  |  | 194 |
|  |  |  |  | 183 |
|  |  |  |  | 195 |
|  |  |  |  | 165 |

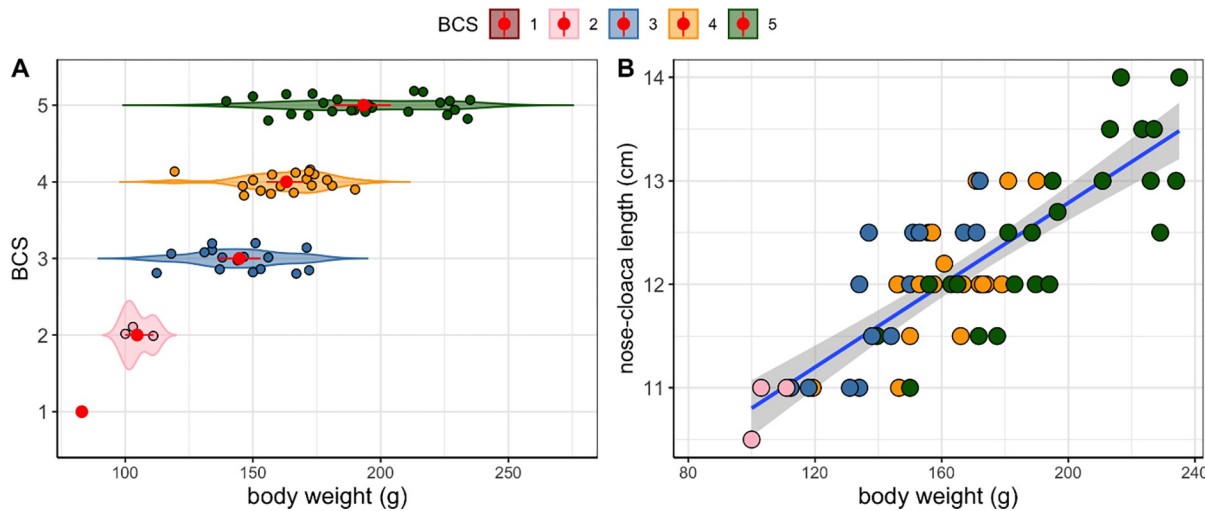

**Fig 4. A.** Violin plot of variance distribution in each BCS class. The violin plot shows the compact distribution of data in each BCS class. Individual data points are jittered within each category. The mean body weight values are also offered as red dots, and the error bars represent 1000-fold bootstrapped 95% confidence intervals. **B.** Scatterplot showing the body weight vs. the nose-cloaca distance color-highlighted by BCS classes. The linear trend in the data ($\beta_{bw}$ = 0.02, SE = 0.002, p<0.001, $R^2_{adj}$ = 0.665) is indicated by the linear regression line with a 95% confidence band.

levels with BCS = 5 as the optimal score. The regression coefficients expressed the average change in body weight per BCS class regarding the intercept level of the model (treatment contrasts). The overall model was significant (F(4,57) = 22.55, p<0.001, $R_{adj}$ = 0.586). Fig 4A shows the variance distribution in each BCS class as violin plots. The 95% confidence intervals overlap between BCS = 4 and BCS = 3, indicating ambiguity in BCS-class attribution using body weight as the sole predictor. A non-significant post hoc test confirmed this ($\Delta\beta_{BCS4}$-$\beta_{BCS3}$ = 18.40, SE = 7.54, p = 0.1197). Therefore, there was no clear body weight difference between BCS classes 4 and 3. The estimates in Table 2 indicate the average body weights per BCS class.

For example, the body weight of BCS = 5 was estimated at $\beta_{Intecept}$ = 193.275 g (SE = 4.481, p<0.001). The body weight of BCS = 4 was significantly lower at 163.067 g ($\beta_{BCL4}$ = -30.208, SE = 6.845, p<0.001), followed by BCS = 3 with 144.669 g ($\beta_{BCL3}$ = -48.606, SE = 7.085, p<0.001) and BCS = 2 with 104.667 g ($\beta_{BCL2}$ = -88.608, SE = 13.443, p<0.001). The largest difference in body weight was observed in BCS = 1 to BCS = 5. Here, a difference of $\beta_{BCL1}$ = -110.275 (SE = 22.404, p<0.001) led to a body weight of 83 g. However, in the BCS = 1 group, only one animal was observed.

The frog nose-cloaca length (NCL) was modeled as a function of the body weight in linear regression. Fig 4B shows the linear regression line of the model with its 95% confidence band. The model was significant (F(1,59) = 119.9, p<0.001, $R_{adj}$ = 0.665), and the average NCL was estimated as 8.82 cm. In addition, the continuous body weight variable contributed

**Table 2. Linear regression model of BCS ~body weight.**

|  | Estimate | SE | t | p-value | stars |
|---|---|---|---|---|---|
| **(Intercept)** | 193.275 | 4.481 | 43.134 | <0.001 | *** |
| bcs4 | -30.208 | 6.845 | -4.413 | <0.001 | *** |
| bcs3 | -48.606 | 7.085 | -6.861 | <0.001 | *** |
| bcs2 | -88.608 | 13.443 | -6.592 | <0.001 | *** |
| bcs1 | -110.275 | 22.404 | -4.922 | <0.001 | *** |

**Table 3. Linear regression model of nose-cloaca length ~bw + bw:BCS.**

|  | Estimate | SE | t | p-value | stars |
|---|---|---|---|---|---|
| **(Intercept)** | 7.661 | 0.446 | 17.186 | <0.001 | *** |
| bw | 0.025 | 0.002 | 10.893 | <0.001 | *** |
| bw:bcs4 | 0.002 | 0.001 | 2.039 | 0.046 | * |
| bw:bcs3 | 0.005 | 0.001 | 3.970 | <0.001 | *** |
| bw:bcs2 | 0.005 | 0.003 | 1.714 | 0.092 | . |

significantly to the model ($\beta_{bw}$ = 0.02, SE = 0.002, p<0.001), corroborating the linear Pearson correlation of body weight and NCL (r = 0.81, $CI_{95\%}$[0.714; 0.887], p<0.001).

Further, the BCS is highlighted in Fig 4B, ranging from pink (BCS = 1) to dark green (BCS = 5). Lower BCSs appeared to relate to shorter body lengths and lower weights. However, the *between*-class transitions were diffuse. Therefore, the linear model was expanded by the bw:BCS interaction to estimate the BCS's independence from body weight (Table 3)

The model with the interaction term was significant (F(4,56) = 40.46, p<0.001, $R_{adj}$ = 0.725) and showed a significant body weight coefficient ($\beta_{bw}$ = 0.025, SE = 0.45, p<0.001). Additionally, the interactions for bw:BCS4 ($\beta_{bw:BCS4}$ = 0.002, SE = 0.001, p = 0.046) and bw:BCS3 ($\beta_{bw:BCS3}$ = 0.004, SE = 0.001, p<0.001) were significant, indicating interdependence from both variables, body weight and the BCS. The interaction of bw:BCS2, however, was not significant at the p≤0.05 threshold but at the p≤0.1 level ($\beta_{bw:BCS2}$ = 0.005, SE = 0.003, p<0.0921), still providing evidence for the interaction of body weight and BCS with lower sample sizes ($n_{BCS2}$ = 3). The significant bw:BCS interaction term requires both variables, body weight, and BCS, to model the NCL.

Finally, the body weight and interaction models were compared in a likelihood-ratio test to evaluate whether the more complex model adds any benefit. The model with the bw:BCS interaction is more complex with 6 degrees of freedom, compared to the model with body weight only (df = 3). Adding the interaction term leads to a significantly better-performing model ($X^2$ = 15.21, p = 0.002), so the hypothesis of restricting the model to a more straightforward design can be rejected.

## Discussion

The first Body Condition Score (BCS) for adult female *Xenopus laevis* was developed and introduced in this study. This BCS was intended to be used for severity assessment in animal husbandry and as an additional parameter in the preparation of protocol-specific score sheets [46] and health examination of adult female *Xenopus laevis*.

With increasing age (≥20 years) [14], however, egg quality decreases [15], so that on the one hand, the limited productive life of the animals in the trials is justified. In addition, the animals are used only a limited number of times for egg production to not exceed a moderate degree of severity. Therefore, biological age is probably not generally reached in the experimental use of clawed frogs. However, since only body length and not the age of the animals are decisive for the purchase by the breeder, no predictions can be made regarding the impact of age on body weight and BCS in the present study.

Hence, the BCS represents the physical status in relation to the individual's body size, independent of other factors. In previously published body condition scores of other species, high score values are usually associated with obesity and thus possible pathological overconditioning of the animals [47, 48]. These body condition score systems distinguish between emaciated and obese animals and therefore define a mid-level as well-conditioned [31, 32] or even as

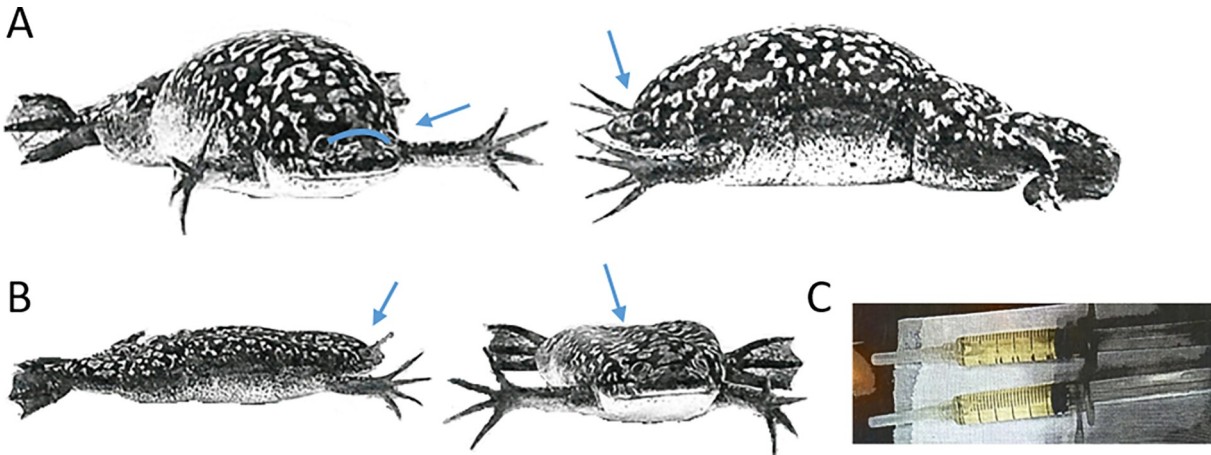

**Fig 5.** High-grade lymph sac edema prior (A) and post puncture (B) of the same specimen, (C) drained clear punctate in syringes. Difference between BCS 5 and pathological lymphatic sac edema: Lymphatic sac edema is indicated by an apparent retraction of the skin between the eyes (see blue marks upper row); skin on the back and thighs shows wave-shaped movements when the frog in question swims due to the edema fluid; the frog swims more sluggishly; no impairment of food intake; clear secretion is drawn off during puncture, and lymph sac collapses again immediately (see blue arrows lower row).

optimum [34, 40, 49]. Our data indicate that in *Xenopus laevis* the BCS 4 and 5 are above a physiological optimum but are not associated with a pathological or avoidable condition and thus represent the desirable condition. This is justified by the fact that due to their particular anatomy and adaptive (ectothermic) metabolism, the animals should be physically resilient to withstand experimental severity (e.g., induced laying of eggs, surgical ovariectomy) [41, 50, 51]. However, care must be taken to distinguish appropriate disease symptoms that cause a biased body shape from the expected condition, e.g., lymph sac edema (Fig 5). The lymphatic sac edema is a pathological clinical phenomenon in clawed frogs [52], which must be distinguished from healthy, obese animals. It should be taken into account that in the dorsal view, the body shape of animals of BCS 5 may be similar to the body shape of frogs with edema of the dorsal lymph sac.

This assumption is justified by the fact that the body composition, ratio, and anatomical position of fat and muscle mass differ from that of mammals. There is no palpable subcutaneous fat tissue which allows conclusions to be drawn about the general nutritional status [41]. Instead, in this species, a large digitiform fat body exists inside the body cavity (coelomic cavity) and can, therefore, neither be assessed visually nor palpated during the examination. At this point, however, it is impossible to determine if the dimension of the fat body correlates with the actual body weight and which size the fat bodies of the animals of different BCS groups have. This would require imaging techniques (e.g., CT or ultrasound) or at least surgical resection of the fat body, which is impossible in the context of the experiments presented here. Another assessment criterion for the physical health condition is measuring the actual body weight and dimensions of the individuals. Both factors can fluctuate significantly due to the intervention (egg laying, surgery, health status, etc.), so this parameter should be evaluated as far as possible during the assessment and always only in combination with the BCS. Depending on the method used to obtain the oocytes/eggs (induction by injection or surgical removal of the ovary), this may affect the animal's body weight. However, during the BCS assessment, these interventions do not affect body length or hindlimb fleshiness (Figs 1 and 2). Since this intervention was performed in all animals evaluated here, this systematic error can thus be neglected. The absolute body weights of the animals (ranging from 83 g to 235 g)

showed that the different allocations of the BCS are associated with varying classes of weight in each case (Table 1). Nevertheless, individual animals belong to the lower or upper limit range or could be classified in a different body condition group merely due to their weight (Fig 3).

Considering the determined values, 33% of the animals show a BCS <3, which does not correspond to the desired optimum. This can be explained by the factors mentioned above (time since the last spawning, husbandry conditions, metabolic activity, and age). In case there are hardly any animals with BCS 4 or 5 in husbandry, an optimization of the housing conditions and a detailed evaluation of the animal population (health monitoring, age, and convalescence phases after interventions) are urgently recommended.

In the performing facility, animals with a BCS 2 are separated, fed, and their weight is monitored strictly (3 times per week). Animals showing a BCS 1 are excluded from any experiment (humane endpoint).

However, assessing a clawed frog with the presented body condition score must always be carried out from two optical perspectives: A lateral view must supplement the information from the dorsal view. This is because breathing and the motion of the hind legs can affect the body's contour. While the external body shape can be easily distorted by inhalation (ventilation of the lungs increases the convexity of the dorsal contour), the extension of the hind limb elongatess the body shape and its body shape appears less pear-shaped. Therefore, the animals are assessed freely-moving, motionless, and relaxed in a water basin to assess BSC values.

It is essential to know that neither the absolute body weight of an animal nor its body dimensions are sufficient to assess the condition of a clawed frog as a single evaluation criterion. These must always be obtained and interpreted in relation to the BCS of the individual.

## Conclusion

It could be demonstrated that the different allocations of the BCS are associated with varying classes of body weight in each case (Table 1). Nevertheless, individual animals belong to the lower or upper limit range or could be categorized in a different body condition group merely due to their weight (Fig 4). The Body Condition Score is therefore intended as a simple, objective tool for personnel involved in experiments and animal caretaker staff to assess conditional changes in *Xenopus laevis* in chronic experiments (e.g., with a focus on oocyte collection) or even in mere animal husbandry, and can classify the physical condition of the animals in general. In addition, this assessment tool tries to define humane endpoints for this underestimated species based on images, to avoid unnecessary animal suffering (experimental refinement) (Fig 3). Since the interaction term of bw:BCS is significant and improves the model, we found evidence for the usefulness of measuring both the body weight and the BCS. Therefore, it is essential to consider body weight, body shape, and condition when assessing *Xenopus laevis*. However, what effect the age of the animals has on the occurrence of the respective BCS groups and what influence the BCS might have on the quality of oocytes and eggs cannot be addressed at this time. Therefore, the BCS is seen as a purely descriptive tool for daily animal care and routine examination of individual animals within laboratory animal husbandry. It also remains to be investigated to what extent the BCS for female Xenopus laevis can be transferred to male animals.

## Supporting information

**S1 Fig. Chart of body condition score.** Chart of Body Condition Score for adult female *Xenopus laevis*.
(DOCX)

**S1 File. Effect of housing on BCS.** Housing conditions were analyzed with linear regression, in which the dependent variable (BCS) was modeled as a function of the housing condition. (DOCX)

## Acknowledgments

We thank Ralf Hausmann, Dominik Wiemuth, Stefan Gründer and Wolfram Antonin for providing the animals. We want to thank the animal caretakers Eva Lotte Kaesbach, Phillip Peters, Leon Heinst, and Martin Heck for their support during the animal examination and Anna Maria Hartmann and Carina Kallen for their support during the data collection.

## Author Contributions

**Conceptualization:** Leonie Tix, René H. Tolba.

**Data curation:** Leonie Tix, Britta Bungardt.

**Formal analysis:** Leonie Tix.

**Investigation:** Leonie Tix, Britta Bungardt.

**Methodology:** Leonie Tix, Lisa Ernst, Gero Hilken, René H. Tolba.

**Project administration:** René H. Tolba.

**Software:** Steven R. Talbot.

**Supervision:** René H. Tolba.

**Validation:** Leonie Tix, Steven R. Talbot.

**Visualization:** Leonie Tix, Steven R. Talbot.

**Writing – original draft:** Leonie Tix.

**Writing – review & editing:** Leonie Tix, Lisa Ernst, Britta Bungardt, Steven R. Talbot, Gero Hilken, René H. Tolba.

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
