## [Decision Letter · Decision Letter 0]

4 Oct 2022

PONE-D-22-22653Establishment of the Body Condition Score for adult female Xenopus laevisPLOS ONE

Dear Dr. Tolba,

Thank you for submitting your manuscript to PLOS ONE. After careful consideration, we feel that it has merit but does not fully meet PLOS ONE’s publication criteria as it currently stands. Therefore, we invite you to submit a revised version of the manuscript that addresses the points raised during the review process. Both reviewers have suggestions for improvements to the manuscript that are reasonably minor.   Please address them point by point.   

We look forward to receiving your revised manuscript.

Kind regards,

Michael Klymkowsky, Ph.D.

Academic Editor

PLOS ONE

Journal Requirements:

Reviewers' comments:

Reviewer's Responses to Questions

**Comments to the Author**

1. Is the manuscript technically sound, and do the data support the conclusions?

Reviewer #1: Yes

Reviewer #2: Yes

2. Has the statistical analysis been performed appropriately and rigorously? 

Reviewer #1: Yes

Reviewer #2: Yes

3. Have the authors made all data underlying the findings in their manuscript fully available?

Reviewer #1: Yes

Reviewer #2: Yes

4. Is the manuscript presented in an intelligible fashion and written in standard English?

Reviewer #1: Yes

Reviewer #2: Yes

5. Review Comments to the Author

Reviewer #1: This manuscript outlines a body condition score for Xenopus frogs along similar lines as those used for other research organisms. There are descriptions of the five score levels ranging from BCS1 (emaciated) to BCS5 ('obese') and a statistical analysis of the relationships of these scores to measurable parameters like length and weight (although these could not be used as single measurements of health). Overall, the body condition scoring should be a useful measure for investigators and staff to assess frog colony health and the effect of repeated oocyte/egg isolation on the animals and could improve objective health assessment.

I have only a few suggestions for improvement:

1. Since the authors used animals from three different housing systems, it might be useful to include this information in the counts of weights and BCS and whether there was any effect of housing on BCS (maybe include housing origin in Table 1?). The small numbers of BCS1-2 animals might make any conclusions difficult, but there might be useful information on any differences in weight or BCS3-5 across the housing conditions.

2. Other organisms often have a unified chart showing the BCS scores next to the views describing the condition criteria. It would be good to include such a chart for people to print out and refer to (maybe as a supplemental figure). Maybe also include in the score descriptions terminology found in other charts "underconditioned", "well-conditioned", etc.

3. Minor typos/word usage suggestions:

page 9:

line 41; check common name use for X. tropicalis (usually "western clawed frog").

line 48; repeat use of term 'undemanding'

line 54; ... live 'purely aquatically'

page 21:

line 295; 'was intended' to be used ....

page 22:

line 333; 'coelomic cavity'

Reviewer #2: The authors provide information regarding body condition score for Xenopus which has not been done previously. The results are well presented but are lacking a few key points that would improve the article for the Xenopus community.

1. The age of the frogs is not presented. The authors do not state if the frogs are all the same age and from the same clutch. How different are the ages and are they all siblings? This information would be helpful.

2. Since Xenopus females are used for oocyte and eggs how does the BCS correlate to the quality of oocytes and eggs? Does BCS impact the quality? Any effect on in vitro fertilization. This information would make BCS score more appropriate for the research community. As it stands the current BCS information would only pertain to those keeping the frogs as pets.

3. What about male frogs?

Overall, the paper is sound but the results are not really of great interest to researchers without more information to correlate to egg/oocyte quality.

6. PLOS authors have the option to publish the peer review history of their article (what does this mean?). If published, this will include your full peer review and any attached files.

Reviewer #1: **Yes: **Douglas W. Houston

Reviewer #2: No

---

## [Author Response · Author response to Decision Letter 0]

18 Nov 2022

Response letter PONE-D-22-22653

Dear Michael Klymkowsky,

First of all, we would like to thank the Editors and the Reviewers for their valuable input and their pro bono work. 

We tried to meet the suggestions of the reviewers and hope that this improved the quality of our manuscript, accordingly. In addition, our response to the individual comments of the reviewers are provided in this point-to-point response letter. 

Referee(s)' Comments to Author:

Reviewer #1: 

1. Since the authors used animals from three different housing systems, it might be useful to include this information in the counts of weights and BCS and whether there was any effect of housing on BCS (maybe include housing origin in Table 1?). The small numbers of BCS1-2 animals might make any conclusions difficult, but there might be useful information on any differences in weight or BCS3-5 across the housing conditions.

We thank you for this valuable feedback and appreciate the comment. For this purpose, we have compared the three different housing conditions and deposited the corresponding data in the supplemental of the manuscript.

At this point, we would like to point out again that the husbandry conditions per se, were not the focus of the paper and the influence on the physiology of the animals cannot be addressed here in detail. 

However, to make the data and graphs transparent and comprehensive to the reader, we have provided further explanations in the supplement.

Please see the reference to the supplemental part of the manuscript in lines 240 - 241:

“The distribution of assessed BCS within the different housing conditions and their effects on BCS are provided in the supplemental materials (supplemental 2).”

Please see here what we added to the supplements concerning your feedback:

“In the present study, animals of the following three different housing conditions were evaluated: large circulating, semi-closed circulation system (17 tanks) with water preparation and conditioning to 21°C ± 1°C (circulation large); small circulating, semi-closed circulation system (4 tanks) with water preparation and conditioning to 20°C ± 1°C (circulation small), individual fresh water tanks (240l tank) without water preparation at 12°C ±3°C (cold water flow-through).

The housing conditions were analyzed with linear regression, in which the dependent variable (BCS) was modeled as a function of the housing condition. The “cold water flow-through” factor was defined as the intercept level in the regression analysis.

Table 4. Result of the linear regression.

The linear regression shows that the “tempered circulation system small” significantly impacts the BCS. In comparison to the “cold water flow-through”, the BCS is 0.88 points lower on average (βsmall=-0.88, SE=0.3, p=0.004), while the “circulation large” shows no difference to the cold water system.

Since the linear regression showed significant coefficient effects, the corresponding ANOVA was also significant (F(2,59)=4.75, p=0.012). Subsequent post hoc tests revealed more specific between-housing condition differences. 

The post hoc test shows that the “circulation small” vs. “circulation large” group was not significant (padj=0.08). However, the difference in average BCS was substantial (∆=-0.64), and there was some evidence of a potential effect (p<0.1). While the comparison of “cold water” vs. “circulation large” was not significant (p=0.69) and showed a much lower difference in the average BCS (∆=0.24), the “cold water flow-through” system showed a significant BCS difference (∆=0.88, p=0.01). The post hoc tests were multiplicity-adjusted with the Tukey method.

2. Other organisms often have a unified chart showing the BCS scores next to the views describing the condition criteria. It would be good to include such a chart for people to print out and refer to (maybe as a supplemental figure). Maybe also include in the score descriptions terminology found in other charts "underconditioned", "well-conditioned", etc.

We are very grateful for the positive review of our manuscript and addressed your remarks accordingly. Therefore, we attached an overview chart of the BCS to the paper's new version as a supplementary figure, enabling working groups to make further use of it. We have also addressed your suggestion and modified the description about the terminology used in previously published BCSs.

Please see the revised manuscript lines:

Line 177-178: “…and a BCS 5 is regarded as a very well-conditioned physical state.”, 

Line 180-181: „…The following assessment criteria are distinguished and defined for the respective BCS levels (Supplemental figure 1).“, 

Line 187: “Frog underconditioned (-): …” and 

Line 197: “Frog very well-conditioned (+++): …” 

Supplemental figure 1: Chart of Body Condition Score for adult female Xenopus laevis

3. Minor typos/word usage suggestions:

Thank you for pointing out typos in the manuscript, which we have corrected, accordingly. Please see the revised version of the manuscript in the appropriate places.

line 42; common name used for X. tropicalis changed to "western clawed frog".

line 50; repeat use of term 'undemanding' changed to 'unpretentious'

line 56; ... live 'purely aquatically'

line 305; 'was intended' to be used ....

line 343/344; 'coelomic cavity'

Reviewer #2: 

1. The age of the frogs is not presented. The authors do not state if the frogs are all the same age and from the same clutch. How different are the ages and are they all siblings? This information would be helpful.

We apologize that this information was not sufficiently addressed in the paper. We have tried to obtain appropriate data and breeding information from the breeder. Unfortunately, they do neither record the individual age, date of spawning nor any relationship of the animals. Therefore, unfortunately, no further information can be given. We have indicated this in line 146/147 of the manuscript.

“However, no statement can be made about the age or the degree of relationship of the animals.”

However, we will gladly take up this point for investigations within the scope of subsequent projects on this topic in order not to neglect these possible influencing factors.

2. Since Xenopus females are used for oocyte and eggs how does the BCS correlate to the quality of oocytes and eggs? Does BCS impact the quality? Any effect on in vitro fertilization. This information would make BCS score more appropriate for the research community. As it stands the current BCS information would only pertain to those keeping the frogs as pets.

Thank you for this interesting and important comment. We are also investigating the significance of the BCS for research. The interpretation of the BCS concerning egg and oocyte quality will be the focus of further investigations and independent tests. We will, therefore, pursue this approach with great interest and publish additional data in due time.

Currently, the BCS is seen as a purely descriptive tool for daily animal care and routine examination of individual animals within laboratory animal husbandry. 

As we are aware of this limitation of the BCS, we addressed this topic in lines 395-398 in the revised manuscript: 

“However, what effect the age of the animals has on the occurrence of the respective BCS groups and what influence the BCS might have on the quality of oocytes and eggs cannot be addressed at this time. It also remains to be investigated to what extent the BCS for female Xenopus laevis can be transferred to male animals. “

What about male frogs?

We are also willing to consider this criticism and examine the transferability of the BCS we presented for female animals to males. However, since these animals are much less frequently kept and used for research on eggs and oocytes, this cannot be answered at this stage.

---

## [Decision Letter · Decision Letter 1]

4 Dec 2022

PONE-D-22-22653R1Establishment of the Body Condition Score for adult female Xenopus laevisPLOS ONE

Dear Dr. Tolba,

Thank you for submitting your manuscript to PLOS ONE. After careful consideration, we feel that it has merit but does not fully meet PLOS ONE’s publication criteria as it currently stands. Therefore, we invite you to submit a revised version of the manuscript that addresses the points raised during the review process.

 Please address reviewer number 2's comment in the text; I believe the manuscript can then be dealt with without further review. 

We look forward to receiving your revised manuscript.

Kind regards,

Michael Klymkowsky, Ph.D.

Academic Editor

PLOS ONE

Journal Requirements:

Reviewers' comments:

Reviewer's Responses to Questions

**Comments to the Author**

1. If the authors have adequately addressed your comments raised in a previous round of review and you feel that this manuscript is now acceptable for publication, you may indicate that here to bypass the “Comments to the Author” section, enter your conflict of interest statement in the “Confidential to Editor” section, and submit your "Accept" recommendation.

Reviewer #1: All comments have been addressed

Reviewer #2: (No Response)

2. Is the manuscript technically sound, and do the data support the conclusions?

Reviewer #1: Yes

Reviewer #2: Yes

3. Has the statistical analysis been performed appropriately and rigorously? 

Reviewer #1: Yes

Reviewer #2: Yes

4. Have the authors made all data underlying the findings in their manuscript fully available?

Reviewer #1: Yes

Reviewer #2: Yes

5. Is the manuscript presented in an intelligible fashion and written in standard English?

Reviewer #1: Yes

Reviewer #2: Yes

6. Review Comments to the Author

Reviewer #1: (No Response)

Reviewer #2: The response to my comments was either we don't know or that is the basis for a future study. I don't have much more to add. This is unfortunate since now the article is of limited use to researchers who use Xenopus, though this descriptive study may be of use to animal care facilities who care for the animals but have no idea about egg, oocyte quality that is needed in these animals for biomedical research.

The authors should include the following statement in the manuscript that they wrote in response to my review:

the BCS is seen as a purely descriptive tool for daily animal care and routine examination of individual animals within laboratory animal husbandry.

Also the abstract text should be slightly modified. In line 24-25 sentence in abstract the authors state the aim was to establish the BCS in terms of experimental and housing refinement. the word experimental should be removed since there is no correlation made between BCS score and experimental use of Xenopus.

7. PLOS authors have the option to publish the peer review history of their article (what does this mean?). If published, this will include your full peer review and any attached files.

Reviewer #1: **Yes: **Douglas W Houston

Reviewer #2: No

---

## [Author Response · Author response to Decision Letter 1]

14 Dec 2022

Dear Sir,

Dear Mr. Klymkowsky,

First of all, we would like to thank the Editors and the Reviewers again for their valuable input and their pro bono work. 

We are sorry that our response to Reviewer 2's comments was not sufficient for him and are of course fine with the request to include the suggested passage in the manuscript.

Referee(s)' Comments to Author:

Reviewer #2: 

1. The authors should include the following statement in the manuscript that they wrote in response to my review:

“the BCS is seen as a purely descriptive tool for daily animal care and routine examination of individual animals within laboratory animal husbandry.”

We regret that our response to your comment was not sufficient in view of the reviewer. We followed your advice and therefore included the statement you suggested in the manuscript. 

Please, find the text section to be inserted in lines 392 – 394:

“Therefore, the BCS is seen as a purely descriptive tool for daily animal care and routine examination of individual animals within laboratory animal husbandry.”

2. Also the abstract text should be slightly modified. In line 24-25 sentence in abstract the authors state the aim was to establish the BCS in terms of experimental and housing refinement. The word experimental should be removed since there is no correlation made between BCS score and experimental use of Xenopus.

In addition, we have adjusted the abstract as requested and removed the term “experimental”. However, in order to preserve the laboratory animal science background at this point, we have modified the information accordingly. We hope that this more correct wording is in line with your intentions.

The corresponding section of the abstract now reads as follows (lines 24 - 25):

“The present study aimed to establish a species-specific BCS for clawed frogs in terms of housing refinement in lab-animal facilities.”

---

## [Editor Report · Decision Letter 2]

19 Dec 2022

Establishment of the Body Condition Score for adult female Xenopus laevis

PONE-D-22-22653R2

Dear Dr. Tolba,

We’re pleased to inform you that your manuscript has been judged scientifically suitable for publication and will be formally accepted for publication once it meets all outstanding technical requirements.

Kind regards,

Michael Klymkowsky, Ph.D.

Academic Editor

PLOS ONE
---

## [Editor Report · Acceptance letter]

28 Dec 2022

PONE-D-22-22653R2 

Establishment of the Body Condition Score for adult female *Xenopus laevis*

Dear Dr. Tolba:

I'm pleased to inform you that your manuscript has been deemed suitable for publication in PLOS ONE. Congratulations! Your manuscript is now with our production department. 

Kind regards, 

on behalf of

Dr. Michael Klymkowsky 

Academic Editor

PLOS ONE